# Factors Associated with Cocaine Consumption among Suicide Victim

**DOI:** 10.3390/ijerph192114309

**Published:** 2022-11-02

**Authors:** Luisa Caroline Costa Abreu, Sarah dos Santos Conceição, Delmason Soares Barbosa de Carvalho, Ana Cristina Machado, Amanda Oliveira Lyrio, Elivan Silva Souza, Cauê Silva Souza, Paulo José dos Santos de Matos, Josicélia Estrela Tuy Batista, Juliano de Andrade Gomes, Alexandre Marcelo Hintz, Priscilla Perez da Silva Pereira, Simone Seixas da Cruz, Isaac Suzart Gomes-Filho, Ana Claudia Morais Godoy Figueiredo

**Affiliations:** 1Health Sciences Teaching and Research Foundation’s, Brasília 70710-907, Federal District, Brazil; 2Department of Health of University of Brasilia, Brasília 70910-900, Federal District, Brazil; 3Department of Public Health of the Federal District, Brasília 70390-125, Federal District, Brazil; 4Department of Epidemiology, Federal University of Recôncavo of Bahia, Santo Antônio de Jesus 44430-622, Bahia, Brazil; 5Department of Health, Feira de Santana State University, Feira de Santana 44036-900, Bahia, Brazil; 6Department of Police of the Federal District, Brasília 70610-907, Federal District, Brazil; 7Department of Nursing, Federal University of Rondônia, Porto Velho 76801-059, Rondônia, Brazil

**Keywords:** suicide, cocaine, death certificates, information systems

## Abstract

Cocaine use is an increasingly frequent event, especially in young people, and can cause irreversible consequences, such as suicide. To evaluate the factors associated with cocaine use in the moments preceding to suicide. This is a population-based, cross-sectional, and analytical study conducted in the Brazilian Federal District by researchers from the Department of Health and the Civil Police Institute of Criminalistics. All people who died due to suicide in 2018 were included in the survey. Cocaine use was considered the dependent variable, and robust Poisson regression was performed to estimate the crude and adjusted prevalence ratios and their respective population confidence intervals. In 2018, 12,157 deaths were recorded, of which suicide accounted for 1.56% of all deaths. It was observed that being between 25 and 44 years old, male, and under the influence of alcohol or cannabis, had a strong positive association with cocaine consumption among suicide victims. Males, people with black skin, with lower level of education, with employment, and who were under the effect of the use of cannabis and/or alcohol in the previous hours of death had a higher propensity to consume cocaine immediately before suicide, with a moderate to strong magnitude of prevalence ratio. The findings of this research indicated the need for monitoring, by health services, of people most vulnerable to suicide through the consumption of psychoactive substances.

## 1. Introduction

Suicide is a phenomenon that contemplates the stages of ideation, planning, attempt, and execution of suicide. Overall, suicide mortality has grown by 60% in the last 45 years [1]. In 2016, it was the second cause of death among young people aged 15 to 29 years [1] and the mortality rate for ages 20 to 30 years was 10.8, 9.0, and 11.5 per 100,000 inhabitants in low, middle, and high-income countries, respectively [2]. Suicide attempts may be 10 to 20 times more frequent than suicide [3].

In the Brazilian context, the suicide mortality rate was in the order of 5.8 per 100,000 inhabitants in 2016 [4], with a 10% increase in suicide rates in the population between 15 and 29 years, from 2011 to 2017 [4]. However, for the same period mentioned above, in the Federal District, there was an increase of 35% in the indicators of death by the aforementioned disease [5]. 

Young individuals are those most vulnerable to suicidal behavior. Most of them reported loneliness that plays a role in causing their suicidal thoughts. Additionally, mental health stress increased their anxiety and depression [6,7,8]. Most of them are male and with 8 to 11 years of schooling level [4,9,10]. Black people in this age group are predominantly more susceptible to suicide [11]. The black population is socially vulnerable, with lower resources for access to education, health, leisure, and work [7].

Economically, suicide and its attempts represent a great burden on society, since they demand public resources that could be allocated to other priority situations and result in a loss of human capital. From a global perspective, the average expenditure for medical care for the victim of suicide attempt in the United States in 2013 was $1537, in addition to the indirect cost associated with loss of productive strength, estimated at $3714 [12]. In Brazil, hospital and deaths by suicide expenses exceeded R$35 million between 1998 and 2007 [13].

The suicide is a complex and multifactorial problem, encompassing predictors such as family history, sociodemographic characteristics, abusive affective relationships, and drug consumption [14]. In this perspective, emotional disorders are pointed out as a favorable aspect of suicidal behavior, especially regarding the use of psychoactive substances such as antidepressants, cannabis, and cocaine [15]. Specifically, with regard to cocaine, it is known that its interaction with other drugs, such as cannabis and alcohol, promotes synergistic changes in the combination of substances, whose effect is reflected in the disorder of emotional processing, decreasing the ability to think, plan and execute, promoting a state of euphoria that ends with suicide [16,17].

The increasing worldwide suicide deaths related to cocaine use is a hypothesis that has not been investigated [18,19]. There is a need to improve knowledge about the theme, to prevent this outcome in susceptible populations and elucidate important aspects for health professionals’ practice in coping with suicide and drug abuse, should be emphasized. This article aims to evaluate the factors associated with cocaine use in the moments preceding suicide.

## 2. Materials and Methods

### 2.1. Design and Context of the Study

This is a population-based, cross-sectional, and analytical study conducted with the Mortality Information System database from the Federal District, which consists of 31 administrative regions and has an extension of 5779.9 km^2^. In 2018, there were 12,157 deaths in the state, among these, 12.81% were due to external causes. These include interpersonal violence and self-directed suicide [20].

The Federal District Department of Health possesses a multidisciplinary team duly trained to verify the quality of the Death Certificate and the Mortality Information System’s data. To verify and characterize deaths by suicide in the Federal District, a research partnership was agreed between the Federal District Department of Health and the Federal District Civil Police. This study was approved by the Research Ethics Committee of the Health Sciences Teaching and Research Foundation, according to laws 466/2012 and 580/2018 of the National Health Council.

### 2.2. Eligibility Criteria

The study involved all individuals who died between 1 January and 31 December 2018, in the Federal District, whose underlying cause of death was defined by codes X60 to X84 of the 10th version of the International Classification of Disease—ICD 10 [21], referring to intentional self-inflicted injury, and recorded in the Mortality Information System. Inconclusive cases regarding the characterization of suicide, as well as those who did not undergo toxicological examination, were excluded from the present study. 

### 2.3. Sources and Instruments for Data Collection

Information on deaths was obtained from the following sources: (1) Mortality Information System; (2) Death Certificates; (3) Death investigation forms of the Federal District Department of Health; and (4) Expert reports of the Civil Police Legal Medical Institute.

### 2.4. Outcome

Cocaine use was considered as the dependent variable: the use of the illicit substance that may have caused cognitive and perception alterations in the individual at the time of suicide [19,21]. The presence of cocaine was verified by toxicological examination performed with 50 mL of blood extracted from the cardiac cavities, according to the protocol of the Legal Medical Institute. Values of cocaine metabolites greater than 10 ng/mL were considered a positive result for the test, and ≤10 ng/mL [22] values were defined as negative.

### 2.5. Exposures

Sociodemographic characteristics and suicide-related data were distributed in categories in accordance with previous studies on the topic [4,23]. The following exposure variables were categorized: age group (13–24; 25–44, 45–59 or ≥60 years), sex (female or male), skin color (black and non-black), employment (yes or no), marital status (with or without a partner) and level of education (≥8 years or <8 years). Data inherent to suicide were evaluated: place of death (hospital and health center, residence, public road or others), previous attempts (yes or no), drug use (yes or no)—cannabis, opioid, benzodiazepines and alcohol, season of the year (summer, autumn, winter or spring), method for the suicide (firearm, sharp instrument, hanging, poisoning, drowning or others) and medical assistance (yes or no).

### 2.6. Data Collection Sources and Instruments

Initially, all the information contained in the Death Certificate was inserted into the Mortality Information System. The multidisciplinary team, previously trained, performed the screening to identify cases suspected and confirmed as suicide. Subsequently, the researchers analyzed the data emitted by the health units in which the individual was admitted before death, as well as the records of all examination reports issued by the Institute of Legal Medicine, in order to understand the characteristics related to the suicide and qualify the information recorded by the physician who completed the death certificate. The findings were registered in a standardized form for data collection.

### 2.7. Data Analysis

Descriptive analysis of all categorical variables was performed according to relative and absolute frequencies. In the second stage, bivariate analysis of sociodemographic characteristics and those related to suicide data were performed. Subsequently, through Poisson regression, prevalence ratios (PR) and their respective population confidence intervals (PCI) were estimated with correction factor for the finite population. The final model was adjusted for age, and statistical analysis was processed in STATA (Data Analysis and Statistical Software, Lakeway, Texas, United States of America), version 16, serial number: 301606315062.

## 3. Results

The present study investigated 156 suicides recorded in 2018, that is, all cases of suicide, and the lost data rate for variables was less than 2%. The loss of inclusive cases (which could not be determined that the cause of death was suicide) was 4% (8 inconclusive cases). In relation to cases not submitted to toxicological examination, the loss was 17.89% (34). Thus, only confirmed suicides, with toxicological tests performed, comprised the population of 156 suicides. Most of the victims were between 25 and 44 years of age and were male. There was a higher occurrence of deaths among blacks and browns, as well as among those victims who did not have a partner. Regarding the social condition of the individuals, most of the victims performed some type of work activity, as well as had a level of education with eight or more years of schooling (Table 1).

Regarding the characteristics of death, it was observed that most occurred at home, as well as highlighting that most individuals did not have medical assistance after the suicide. In relation to the previous attempted suicide, it was noted that most of the victims had antecedents. The method adopted for self-inflicted injury with the greatest predominance was hanging. Concerning the seasonality of the event investigated, a greater number of events were observed in the summer.

Toxicological exams showed the presence of alcoholic beverage in the body of 25.32% of the victims, as well as evidence of cannabis in 10.90%. The use of benzodiazepines and opiates was revealed to be a factor in 15.22% and 6.02% of deaths, respectively (Table 2).

Because it is a population-based study, when analyzing the sociodemographic characteristics and related to the death of victims by suicide, according to cocaine consumption, most factors showed an association of moderate to strong magnitude (Table 3). This pattern was observed both in the crude and age-adjusted analysis, except for the marital status variable.

In the adjusted analysis, being a man, of black skin race/color, without occupation of work, with a level of education less than or equal to 8 years of schooling and who were under the effect of the use of marijuana and/or alcohol in the previous hours and/or at the time of death, were more likely to have consumed cocaine before suicide. 

The factors that showed greater strength of association with cocaine use in suicide victims were aged between 25 and 44 years, male gender and presence in the blood of marijuana and alcohol. Among these, men were approximately 3.5 times more likely to use cocaine among suicide victims (adjusted PR: 3.45; PCI: 1.41–8.37) than women. The presence of alcohol in serum was almost 3 times more frequent among people who used cocaine among suicide victims than those without the presence of the cannabis drug (PRadjusted: 2.85; PCI: 1.62–5.02).

## 4. Discussion

The results showed that 15.38% of suicide victims submitted to post-mortem toxicology had cocaine metabolite values higher than 10 ng/mL in their blood. In addition, being a man, of black, young, with a higher level of education and equal to 8 years of schooling and without work occupation were the factors positively associated with cocaine use immediately before suicide. These findings are in agreement with the world, since young adults (15–29 years) are the most exposed group in middle- and low-income countries. In Brazil, studies have shown indicators similar to those found in the Federal District [7,24]. 

Being male was considered an important predictor for cocaine use, confirming the findings of one systematic review [25]. According to a study conducted in a Canadian hospital [26], there was a high prevalence of males (73.2%) and cocaine use (91.5%) in suicide attempts among drug users. However, in a Brazilian survey, between 2011 and 2016, it was shown that females are the most likely to attempt suicide [4,7].

Socially, men are more prone to behavioral deviations, due to the influence of social or cultural issues rather than being a characteristic inherent to the sex. Moreover, they often present impulsive behaviors and use a certain substance to obtain a feeling of well-being and pleasure. Regarding the type of drugs consumed, they are more likely to use cocaine, alcohol, and cannabis earlier, for longer, in quantity, and more frequently than women [27]. 

It was observed during the toxicological investigation that 7.69% of the victims used alcohol and cocaine simultaneously. From this perspective, there was a 2.85 times higher probability of individuals who had consumed alcohol also using cocaine prior to the suicide. The potentiation of compulsive and impulsive behaviors is 10 to 14 times greater, promoted by the interaction between cocaine and alcohol. In a study conducted in Spain [28], 34% of suicides were among those who used the two drugs, compared to 32% and 27% of suicides among exclusive users of alcohol or cocaine, respectively.

The maximization of the aforementioned association results from the process of hepatic transesterification of cocaine, which, via the combination with ethanol, provokes the formation of cocaethylene [29]. Although this metabolite promotes organic repercussions similar to the effect caused by cocaine use exclusively, it has a longer half-life of 2 and a half hours, consequently extending the period of exposure to drug implications, as well as neuropsychiatric symptoms [16]. 

It can be seen that the risk of suicide is potentiated through the concomitant use of cannabis and cocaine [16]. The interaction of both drugs in the human body promotes reward system hyperactivity by combining the synergistic activities of the receptors of the two substances, whose effect is reflected in the disorder of the emotional or learning processing [17]. Finally, it results in the worsening of executive function (ability to think, plan and perform), due to a deficit in inhibiting control and repercussions of dopamine release, promoting a state of euphoria [30].

Black skin presented moderate magnitude in relation to acute cocaine use; this fact can be justified by historical issues, socioeconomic disadvantages, and experiences of discrimination [31]. However, there was divergence from previous studies [18,32], which stated that non-black skin was more prevalent among suicide victims who used drugs. When analyzing employment, the data presented differed from the observations of another research [33] developed in a Brazilian hospital, which highlighted a preponderance of the absence of employment ties to suicide attempts of cocaine users. Unemployment influences both the higher consumption of illicit drugs and the effectiveness of suicide [34].

Regarding the level of education, referring to the years of study, there was a similarity in relation to the evidence of one American investigation [35]. Concerning marital status, the present study results confirmed findings of another investigation [33], which found a higher prevalence in victims with no partner and who used cocaine prior to suicide. It is noteworthy that cocaine use predisposes to stigmas that hinder the socialization of the individual [36]. However, it was not possible to investigate the impact of belonging to groups of greater social vulnerability. 

The data presented here make clear the need for the implementation of suicide prevention strategies, since it is recognized that illicit drug use by youth, especially cocaine, results in the potentiation of suicidal behavior. At the 66th World Health Assembly, an action plan was established from 2013 to 2020, which provides for the adoption of suicide prevention strategies and the improvement of their notification in public record systems [37]. In this way, it aims to improve and train the teams working in health services to properly manage individuals exposed to risk factors in vulnerable groups to minimize cases of suicide [38]. It is suggested the creation of an epidemiological surveillance protocol in health to track individuals with suicidal ideation. In this case, all people with suicidal ideation, cocaine and alcohol users should be monitored by health services.

Among the potential limiting factors of this research, one can mention the loss of information during the collection, due to inequity in the filling out of the death certificate. The possibility of underreporting cannot be ruled out; however, all cases considered inconclusive were reviewed. Another limitation of the study is the distribution of the number of suicides found in the population. Some factors have had their subcategories in values lower than 10, which may explain the lack of precision in some results. Moreover, although this type of study does not commonly allow for evaluating the temporality of events, in the present study, it is possible to establish this relationship between sociodemographic variables and cocaine use, since the individuals die by suicide after the use of the substance.

Regarding the strengths, to ensure the reliability of the observed results, we highlight the data collection and investigation of suicide death performed by trained staff, as well as the use of a standardized form and double verification of information by independent researchers, in order to minimize possible biases of selection and measurement. In an attempt to minimize confounding, robust analytical techniques for studies of finite whole populations were applied, taking into consideration the variability of confidence intervals to interpret the findings. There is a need for studies with longitudinal methods that more rigorously investigate the aspects of the causality of cocaine consumption and suicide.

## 5. Conclusions

Demographic and lifestyle factors were those most strongly associated with cocaine use in suicide victims. It is important to highlight the need to better qualify health services for the generation of more reliable data on suicide and, consequently, improve the quality of health indicators, a tool necessary for the development of more assertive public policies, better planning of preventive actions and minimization of preventable deaths.

## Figures and Tables

**Table 1 ijerph-19-14309-t001:** Sociodemographic characteristics of suicide victims.

**Characteristics**	**n = 156 (%)**
**Age (years)**	
13–24	**43 (27.56%)**
25–44	**68 (43.59%)**
45–59	**25 (16.03%)**
≥60 anos	**20 (12.82%)**
**Sex**	
Female	**51 (32.69%)**
Male	**105 (67.31%)**
**Skin Color**	
Non-black	**64 (41.03%)**
Black	**92 (58.97%)**
**Employment**	
Yes	**136 (87.18%)**
No	**20 (12.82%)**
**Schooling (years) ***	
≥8	**113 (72.90%)**
<8	**42 (27.10%)**
**Marital Status ***	
With partner	**51 (33.12%)**
Without partner	**103 (66.88%)**

Source: Data extracted from the expert reports of the Technical Police Department of the Civil Police. * Lost information.

**Table 2 ijerph-19-14309-t002:** Drug used and death characteristics of suicide victims.

**Characteristics**	**n = 156 (%)**
**Place of Death**	
Hospital and health center	**25 (16.03%)**
Residence	**102 (65.38%)**
Public Road	**16 (10.26%)**
Others	**13 (8.33%)**
**Drug use**	
**Benzodiazepines ***	
Yes	**21 (15.22%)**
No	**117 (84.78%)**
**Alcohol ***	
Yes	**39 (25.32%)**
No	**115 (74.68%)**
**Cocaine ***	
Yes	**24 (15.38%)**
No	**132 (84.62%)**
**Cannabis ***	
Yes	**17 (10.90%)**
No	**139 (89.10%)**
**Opioid ***	
Yes	**8 (6.02%)**
No	**125 (93.98%)**
**Season of the year ***	
Spring	**40 (25.64%)**
Summer	**51 (32.69%)**
Autumn	**42 (26.92%)**
Winter	**23 (14.75%)**
**Method for the suicide ***	
Firearm	**39 (25.00%)**
Sharp instrument	**1 (0.64%)**
Hanging	**86 (55.13%)**
Poisoning	**26 (16.67%)**
Drowning	**1 (0.64%)**
Others	**3 (1.92%)**

* Lost information.

**Table 3 ijerph-19-14309-t003:** Factors associated with cocaine use in suicide victims.

	Cocaine Consumption		PR_(CRUDE)_	PR_(ADJUSTED)_
Characteristics	Yes	No	(PCI)	(PCI)
	n = 24 (%)	n = 132 (%)		
**Age (years)**				
13–24	5 (11.63%)	38 (88.37%)	**1.16** (0.35–3.82)	Ref. ***
25–44	17 (25%)	51 (75%)	**2.50** (0.86–7.19)	_
45–59	0 (0%)	25 (100%)	_	_
≥60 anos	2 (10%)	18 (90%)	Ref. **	_
**Sex**				
Female	3 (5.88%)	48 (94.12%)	Ref. ** **3.40** (1.06–10.87)	- **3.45** (1.41–8.37)
Male	21(20%)	84 (80%)		
**Skin Color**				
Non-black	7 (10.94%)	57 (89.06%)	Ref. ** **1.68** (0.74–3.83)	- **1.61** (0.87–2.98)
Black	17 (18.48%)	75 (81.52%)		
**Employment**				
Yes	19 (13.97%)	117 (86.03%)	Ref. ** **1.78** (0.75–4.25)	- **1.97** (1.01–3.83)
No	5 (25%)	15 (75%)		
**Schooling (years) ***				
≥8	16 (14.16%)	97 (85.84%)	Ref. ** **1.34** (0.62–2.90)	- **1.38** (0.76–2.49)
<8	8 (19.05%)	34 (80.95%)		
Marital Status *				
With partner	7 (13.73%)	44 (86.27%)	Ref. ** **1.20** (0.53–2.71)	- 1.00 (0.47–2.09)
Without partner	17 (16.50%)	86 (83.50%)		
**Drug use**				
**Alcohol ***				
Yes	12 (30.77%)	27(69.23%)	**2.94** (1.44–6.01) Ref. **	**2.85** (1.62–5.02)
No	12 (10.43%)	103 (89.57%)		
**Benzodiazepines**				
Yes	3 (14.29%)	18 (85.71%)	0.98 (0.31–3.06) Ref. **	1.00 (0.43–2.53)
No	17(14.53%)	100(85.47%)		
**Opioid ***				
Yes	1 (12.50%)	7 (87.50%)	0.91 (0.13–6.05) Ref. **	1.00 (1.00–1.02)
No	17 (13.60%)	108 (86.40%)		
**Cannabis ***				
Yes	5 (29.41%)	12 (79.59%)	**2.15** (0.92–5.01) Ref. **	**2.02** (1.03–3.99)
No	19 (13.67%)	120 (86.33%)		

Source: Data extracted from the expert reports of the Technical Police Department of the Civil Police. * Lost information. ** Ref. = Reference. *** Adjusted by age.

## Data Availability

Not applicable.

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
