# Peer review of "Factors Associated with Cocaine Consumption among Suicide Victim"

_ijerph, 2022, doi:10.3390/ijerph192114309_

Round 1

Reviewer 1 Report

1. Introduction: Please explain further why suicide rates among young people aged are increasing.

2. Introduction: Further descriptions of the relationship between cocaine consumption and suicide are needed.

3. Materials and Methods: Please provide the rationale for categorizing variables such as age and education level.

4. Materials and Methods: Please describe the process by which the total number of suicide victims is obtained.

5. Materials and Methods: Please indicate the value of n above all tables.

6. Discussion: Evidence should be provided that men use the drug more.

Author Response

Dear, follow the answers and adjustments as requested by the reviewers. Responses are highlighted in yellow and changes in red.
Regards,

Reviewer 2 Report

This manuscript presents results of mortality database records for all suicide deaths (n = 156) occurring in the Brazilian Federal District during an entire calendar year. The primary purpose of the investigation was to identified demographic and other factors that were associated with cocaine use as determined from toxicological evidence (= 15% of the total suicides). The authors provide a number of such positively associated factors. It is suggested that these findings, related to drug use and suicidal behaviors, support attention to this factor along with the presence of other associated factors in measures toward suicide prevention.

The greatest strength of the study lies in the investigation of (virtually) all suicides in a complete geographic area within a full calendar year. This is a surveillance study. Another strength is the identification of a number of specific characteristics of suicides, both demographic as well as social, that were positively associated with the use of cocaine among those suicides. An additional strength is the methodological procedures employed to provide information about the suicide decedents and derived from other records and the standardization of an information form containing the set of data for each death. Finally, the focus on cocaine consumption in deaths by suicide (particularly in the immediate period before the suicide) has been a neglected issue.

There are some issues and questions that might be addressed.

(1) Sample size and multiple factors and subcategories –The tabled information provides the breakdown of the 156 suicides (age, sex, skin color, employment, drug use, alcohol, method of suicide, etc.). However, the number of categories (e.g., for age there are 4 age groupings) creates issues, that is, the number of cases in cells are relatively small. When looking at the cells for cocaine consumption (24 cases, and 132 without cocaine), however, where associations between cocaine consumption and the other many factors are shown, the number of cases are not infrequently in single digits (i.e., <10). When discussing the limitations of the study, there is no mention of potential problems associated with small numbers of cases included.

(2) Implications –The authors (in the abstract) suggest that their findings show the need for health services to monitor people most vulnerable to suicide through the consumption of drugs such as cocaine. This suggestion seems reasonable and the findings identify several groups, including men, those aged 25-44, presence of marijuana and alcohol (especially both alcohol and cocaine) in the blood. Monitoring, by health services, all those with just these characteristics (among the many others identified) would likely represent a very large number of people and case loads. Practical questions about this implication arise,  such as how are the individuals to be monitored identified and for how long would monitoring be performed? What intervention(s) should be administered?, etc.  Identification of cases who use cocaine, cocaine with alcohol, men, etc. would undoubtedly lead to large numbers of false positives regarding suicide. Would the authors have any practical and/or specific thoughts regarding how to implement such measures?

(3) Miscellaneous issues – (a) It is noted that “inconclusive cases regarding the characterization of suicide” and cases that did not undergo toxicological examination were excluded.  How many deaths for each of these two categories were excluded? How many total suicides were identified? That is, were the excluded cases included in the total of 156 suicides or does this 156 represent only those cases that were deemed suicide AND had toxicology information? These specifics would be useful for clarity within the study of all deaths with an underlying cause of death as suicide in the calendar year of study. (b) The use of “self-extermination” to refer to suicide deaths is quite uncommon. The word “extermination” has connotations that might be undesirable. (c) On page 8 of 12 it is stated that “It was observed during the toxicological investigation that 30.77% of the victims used alcohol and cocaine simultaneously.” There are other locations where the use of “the suicide victims” occurs and it seems that the reference is to the entire set of 156 suicides. But in this page 8 instance it is not possible that it is from among all 156 suicides because it must be only among the 24 cocaine consumption cases (since these deaths had both alcohol and cocaine present). Possible clarification of this seems warranted here and could be checked elsewhere. 

There are some other minor wording/typo suggestions that might be considered:

1.     In the last sentence of page 8 of 12, “In this way, it aims to improve and train the teams working in health services to properly manage individuals exposed to risk factors and inserted in vulnerable  groups to minimize cases of suicide.” The word “insert” is not clear.

2.     On page 9 of 12, line 256, the wording “committed suicide” occurs. Though existing in the long-term literature on suicide, “committed” is discouraged in contemporary suicidology and is seen as a pejorative word and one that is harmful to the surviving family of those who have died by suicide. Perhaps “die by suicide” would be one possible substitute.

3.     On line 72, page 2 of 12, the word “invested” appears.  It would seem that what is intended there is the word “investigated”.

Author Response

(The authors gave the same response as above.)

Round 2

Reviewer 2 Report

This is a population study of deaths by suicides who also used cocaine. The study results provide basic evidence identifying demographic subgroups of the population among which suicide and cocaine occur together. The results have possible implications that may be of use to health and mental health professionals in monitoring and possibly intervening to prevent suicidal behaviors.